**Exploring the ocean mesoscale at reduced computational cost with FESOM 2.5: efficient**
**modeling strategies applied to the Southern Ocean**
Nathan Beech[1], Thomas Rackow[2], Tido Semmler[3], and Thomas Jung[1,4]
1. Alfred Wegener Institute Helmholtz Center for Polar and Marine Research, Bremerhaven,
Germany
2. European Center for Medium-range Weather Forecasts, Bonn, Germany
3. Met Eireann, the Irish Meteorological Service, Dublin, Ireland
4. Department of Physics and Electrical Engineering, University of Bremen, Bremen, Germany
Corresponding Author: Nathan Beech (Nathan.beech@awi.de)
**Abstract**
Modeled projections of climate change typically do not include a well-resolved ocean mesoscale
due to the high computational cost of running high-resolution models for long time periods. This
challenge is addressed using efficiency-maximizing modeling strategies applied to 3 km simulations of
the Southern Ocean in past, present, and future climates. The model setup exploits reduced-resolution
spin-up and transient simulations to initialize a regionally refined, high-resolution ocean model during
short time periods. The results are compared with satellite altimetry data and more traditional eddy-
present simulations and evaluated based on their ability to reproduce observed mesoscale activity and to
reveal a response to climate change distinct from natural variability. The high-resolution simulations
reproduce the observed magnitude of Southern Ocean eddy kinetic energy (EKE) well, but differences
remain in local magnitudes and the distribution of EKE. The coarser, eddy-permitting ensemble simulates
a similar pattern of EKE, but underrepresents observed levels by 55%. At approximately 1 °C of
warming, the high-resolution simulations produce no change in overall EKE, in contrast to full ensemble
agreement regarding EKE rise within the eddy-permitting simulations. At approximately 4 °C of
warming, both datasets produce consistent levels of EKE rise in relative terms, although not absolute
magnitudes, as well as an increase in EKE variability. Simulated EKE rise is concentrated where flow
interacts with bathymetric features in regions already known to be eddy-rich. Regional EKE change in the
high-resolution simulations is consistent with changes seen in at least four of five eddy-permitting
ensemble members at 1 °C of warming, and all ensemble members at 4 °C. However, substantial noise
would make these changes difficult to distinguish from natural variability without an ensemble.
**Plain Language Summary**
Cost-reducing modeling strategies are applied to high-resolution simulations of the Southern
Ocean in a changing climate. They are evaluated with respect to observations and traditional, lower-
resolution modeling methods. The simulations effectively reproduce small-scale ocean flows seen in
satellite data and are largely consistent with traditional model simulations after 4 °C of warming. Small-
scale flows are found to intensify near bathymetric features and to become more variable.

## 1 Introduction

Mesoscale activity in the Southern Ocean has been the subject of much research and interest in recent years due to the intensification of Southern Hemisphere westerlies (Marshall, 2003), the phenomena of eddy saturation and compensation (Munday et al., 2013; Bishop et al., 2016), and the potential for carbon sequestration in the face of ongoing anthropogenic emissions (Sallée et al., 2012; Landschützer et al., 2015; Frölicher et al., 2015). Satellite observations already reveal an intensification of eddy activity in the Antarctic Circumpolar Current (ACC) and changes are attributed primarily to wind stress (Marshall, 2003; Hogg et al., 2015; Martínez-Moreno et al., 2021). Modeling studies have been able to reproduce the observed changes, as well as project continued intensification throughout the $21^{st}$ century (Beech et al., 2022), but the modeled results rely on only partially resolved eddy activity relative to observations, leaving open the possibility for new findings or greater clarity.

Advances in computational capabilities have enabled ocean modeling science to make great progress in overcoming the substantial computational burden of simulating the mesoscale. However, shortcomings remain, particularly in the Southern Ocean where the Rossby radius can be as small as 1 km, increasing the computational cost of resolving eddies (Hallberg, 2013). Even model resolutions that can generally be considered eddy-resolving are only eddy-permitting poleward of 50° if grid spacing does not vary in space (Hewitt et al., 2020). This highlights an efficiency challenge in simulating the mesoscale with traditional model grids; resolutions necessary to resolve high-latitude, small-radius eddies are both prohibitively expensive and unnecessary to resolve mesoscale eddies in the lower latitudes. Fortunately, a growing number of modeling alternatives to traditional grids now enable dynamic spatial allocation of resources (Danilov, 2013; Ringler et al., 2013; Danilov et al., 2017; Jungclaus et al., 2022), creating the opportunity to more efficiently resolve the mesoscale.

As resource allocation in high-resolution modeling becomes spatially flexible in the pursuit of more efficient configurations, the temporal component must also be scrutinized for efficiency. Traditional modeling approaches require long spin-up periods in order to equilibrate the deep ocean and reduce

model drift (Irving et al., 2021). Although the impacts of drift are not negligible, they generally affect
large-scale processes in the deep ocean; mesoscale processes that require high resolutions to simulate are
typically fast-to-equilibrate and will appear relatively quickly wherever large-scale ocean conditions lead
to their creation. Admittedly, one cannot entirely disentangle the two scales, as mesoscale activity does
affect the position of fronts, stratification, and the paths of ocean circulation (Marshall et al., 2002;
Marzocchi et al., 2015; Chassignet and Xu, 2017). Yet, with equilibration times for the deep ocean on the
scale of thousands of years (Irving et al., 2021), the possibility, and ultimately necessity, to reduce the
resolution of spin-up runs relative to production runs must be investigated.

Advancing the concept of dynamic temporal allocation of resources further, the traditional

transient climate change simulation also represents an efficiency bottleneck for some applications; by
modifying the climate continuously in time, each year of a transient simulation is effectively a single
realization of a global mean climatic state that varies from the following and preceding years by only a
fraction of a degree. For some applications, like hindcasts of real events or trend analysis, this approach
may be desirable, but for assessing the impacts of climate change with limited resources and a low signal-
to-noise ratio, a larger sample of realizations for a consistent climatic state may be more suitable.

Aside from oceanic concerns, the atmosphere can have substantial impacts on mesoscale activity

in climate models. Most simply, with a coupled atmosphere, absolute surface winds will react to ocean
eddy activity, whereas atmospheric forcing will not, resulting in more eddy killing by wind stress
(Renault et al., 2016). Additionally, an atmosphere coupled to a high-resolution ocean must be of
similarly high resolution for certain mesoscale interactions to be resolved (Byrne et al., 2016). Ultimately,
the modeled atmosphere further escalates the already exponential cost of increasing ocean resolution by
requiring more computational resources in order for the benefits of the resolved mesoscale to fully
transfer to the broader climate.

To address the computational inefficiencies outlined above, a novel simulation configuration is

proposed, combining several experimental modeling approaches. Simulations will exploit the multi-
resolution Finite volumE Sea-ice Ocean Model (FESOM) (Danilov et al., 2017) employing a high-
resolution unstructured mesh that concentrates computational resources on the Southern Ocean, while
maintaining grid resolution in the remainder of the global ocean that can still be considered high-
resolution, as in, for example, HighResMIP (Haarsma et al., 2016). The multi-resolution strategy
overcomes the efficiency challenges of resolving high-latitude eddies without needlessly increasing
tropical resolutions, as well as limiting the focus and computational requirements to one hemisphere. The
high-resolution simulations will make use of a spin-up simulation on a medium-resolution, eddy-
permitting mesh to avoid the computational burden of allowing an eddy-resolving ocean to equilibrate
deep, slow-changing processes. The eddy-permitting mesh will also be used to simulate the transient
periods between shorter, high-resolution time slices, increasing the signal-to-noise ratio of the results by
separating the production data further in time and the progression of anthropogenic climate change.
Finally, the ocean model will be forced with atmospheric data from existing coupled simulations
(Semmler et al., 2020). Although this will not facilitate mesoscale atmosphere-ocean interaction, the
simulation will reflect the climatic development of an eddy-permitting simulation of the future
atmosphere without the additional computational requirements.

The Southern Ocean is one of the world's hotspots for mesoscale activity and a region where

substantial change is anticipated in the context of anthropogenic climate change (Beech et al., 2022).
Simultaneously, the high latitude of the region makes eddy-resolving model simulations computationally
demanding and observational data relatively scarce (Auger et al., 2023; Hallberg, 2013). Yet, as the
climate changes, the importance of the Southern Ocean grows as a heat and carbon sink, an ecosystem,
and a medium for feedback between the atmosphere and ocean (Byrne et al., 2016; Frölicher et al., 2015).
Thus, the study of the Southern Ocean demands innovation in the modeling field to produce high-
resolution simulations at reduced computational cost. This study maximizes grid resolution relative to
computational cost using an unstructured, multi-resolution grid, a medium-resolution spin-up simulation,
and atmospheric forcing from lower-resolution coupled simulations in order to focus resources as much as
possible on resolving mesoscale activity in the study region. The resulting simulations enable an
exploratory analysis of the past, present, and future of the Southern Ocean with a fully resolved
mesoscale. Simulations with this cost-efficient, high-resolution configuration are presented in comparison
to a comprehensive ensemble of eddy-permitting simulations to assess the performance of the efficiency-
focused approach in reproducing mesoscale activity and its response to climate change.

## 2  Methods

### 2.1 Experimental setup

This analysis contrasts a subset of simulations from AWI-CM-1-1-MR's contribution to the sixth

phase of the Coupled Model Intercomparison Project (CMIP6; Semmler et al., 2020), (hereafter referred
to as the AWI-CM-1 ensemble) with single-member stand-alone ocean simulations using an updated
version of FESOM (FESOM 2.5) and a mesh substantially refined to a resolution surpassing 3 km in the
Southern Ocean (hereafter referred to as the SO3 simulations) (Supplementary Figure 1). Observations of
ocean surface velocity derived from satellite altimetry data are also used to evaluate model performance
for both modeled datasets during the period of overlap with the altimetry record. The AWI-CM-1
simulations consist of the five-member ensemble of historical simulations and the five-member ensemble
of climate change projections under shared socioeconomic pathway (SSP) 3-7.0 which were performed by
AWI-CM-1-1-MR in CMIP6 (Semmler et al., 2020). These are state-of-the-art CMIP6 experiments and
benefit from the multiple ensemble members and long spin-up times that CMIP simulations typically
boast. However, while the AWI-CM-1 ensemble reproduces eddy activity remarkably well within the
context of CMIP6 (Beech et al., 2022), high-resolution ocean modeling now far surpasses even the
highest ocean resolutions in the CMIP6 ensemble. Conversely, the SO3 simulations push the limits of
ocean resolution but rely on several measures for maximizing computational efficiency that may impact
the robustness of the simulations. Details on the experimental setup for CMIP6 and ScenarioMIP are
widely available (Eyring et al., 2016; O'Neill et al., 2016) and information more specific to AWI-CM-1-
1-MR's contribution has been published previously (Semmler et al., 2020). The following sections will
outline the details of the SO3 simulations.

To produce initial conditions for the high-resolution model simulations on the SO3 mesh, a

medium-resolution, eddy-permitting, ocean-only transient simulation was first run from 1851 to 2100
using the same ocean mesh employed by AWI-CM-1-1-MR in CMIP6 (Semmler et al., 2020). This mesh
has been shown to effectively reproduce eddy activity in active regions while maintaining a
computational cost comparable to a traditional ¼ ° model (Beech et al., 2022). The transient simulation
was initialized with conditions for ocean temperature and salinity, as well as sea ice concentration,
thickness, and snow cover taken from the end of the first year (1850) and first ensemble member
(r1i1p1f1) of AWI-CM-1-1-MR's historical simulations in CMIP6 (Semmler et al., 2018, 2020, 2022a,
b). In this way, the model undergoes a semi-cold start in which ocean conditions are not exact
continuations of the previous coupled simulation, but should be far closer to equilibrium than a true cold
start initialization. The eddy-permitting transient simulation was forced using atmospheric data from the
same ensemble member of the historical CMIP6 simulations until 2014 (Semmler et al., 2022a), and
thereafter using the first ensemble member of AWI-CM-1-1-MR's ScenarioMIP simulations for SSP 3-
7.0 (Eyring et al., 2016; O'Neill et al., 2017; Semmler et al., 2022b). This approach to forcing takes
advantage of a coupled simulation, CMIP6, to produce a forcing dataset of better temporal and spatial
coverage than the observational record and which maintains a realistic transient climate throughout
anthropogenic impacts during the 21$^{st}$ century.

In the years 1950, 2015, and 2090, FESOM is reinitialized with the high-resolution ocean grid,

SO3 (Supplementary Figure 1), using the same semi-cold start approach and forcing dataset that was
implemented for the eddy-permitting transient simulation described previously. These years were chosen
to represent a historical period, beginning in 1950, when the effects of climate change on EKE should be
small or none (Beech et al., 2022); a near-present period, beginning in 2015, in which the simulations will
overlap with satellite altimetry data; and a projected period, beginning in 2090, which should include a
strong climate change signal. The latter two simulated periods represent 1.07 °C and 3.74 °C of warming,
respectively, in the first ensemble member of the AWI-CM-1 ensemble defined as a rise in the 21-year
running mean of global mean two-meter air temperatures. Warming of the ensemble mean is similar: 1.08
and 3.76 °C respectively, and warming is henceforth approximated as 1 °C and 4 °C in Fig. 4 and the text.
Initial conditions for these shorter time-slice simulations are taken from the end of the previous year of
the eddy-permitting transient simulation. The high-resolution simulations are each integrated for six years
with the first year ignored as a true spin-up, leaving five years of data for each time period. The high-
resolution grid is, in truth, a regionally refined mesh in which a 25 km global resolution is refined to
approximately 2.5 km, following Danilov (2022), primarily south of 40 °S, but with other pertinent
regions, such as the Agulhas Current and several narrow straits, also refined. In this way, the model is
able to simultaneously achieve eddy-rich conditions in the Southern Ocean and many of the nearby active
regions, as well as a global resolution that would still be considered high in the context of CMIP6
(Hallberg, 2013; Hewitt et al., 2020). While model drift may be a concern with such a short true spin-up
period, this should affect each of the high-resolution time slices similarly and to a limited extent due to
their short integration lengths. Thus, the differences between the high-resolution ocean simulations should
primarily reflect anthropogenic climate impacts simulated during the eddy-permitting transient run and
present in the forcing dataset.
**2.2 Model configuration**

The Finite volume Sea-ice Ocean Model version 2.5 is a post-CMIP6 era model, having been

refactored to a finite-volume configuration from the finite-element version (FESOM1.4, Q. Wang et al.,
2014) employed in CMIP6, and transitioned to arbitrary Lagrangian Eulerian vertical coordinates, among
other improvements (Danilov et al., 2017; Scholz et al., 2019, 2021). FESOM's most distinguishing
feature among mature ocean models is the unstructured horizontal grid that exploits triangular grid cells
which can smoothly vary in size to change the horizontal grid resolution in space. In these simulations,
full free surface, or z*, vertical coordinates were used, allowing the vertical model layer thicknesses to
change in time. Gent-McWilliams eddy parameterization (Gent and McWilliams, 1990) is scaled with
resolution according to Ferrari et al. (2010) and vertical mixing is simulated by a *k*-profile
parameterization scheme (Large et al., 1994).

The SO3 mesh consists of over 22 million surface elements (triangle faces) or 11 million surface

nodes (triangle vertices) and 70 vertical layers. The simulations produce about 1.1 terabytes of data per
year of 3D data stored on nodes. For reference, the medium-resolution mesh used in the AWI-CM-1
ensemble is 1.6 million surface elements or 0.83 million surface nodes and 46 vertical layers and
produces approximately 56 GB per year of 3D data stored on nodes. The model was run on 8192 CPU
cores and with a typical throughput of approximately 0.65 simulated years per day, consuming
approximately 5.5 million CPU hours in total despite the various cost-saving modeling approaches. It
should be noted, however, that the throughput in high-resolution production simulations like this is highly
dependent on the volume and choice of data being saved. The simulations and following analysis were
performed using the high-performance computing system, Levante, at the German Climate Computing
Center (DKRZ).

The ocean model is forced by several atmospheric variables at a six-hour resolution, although one

forcing variable, humidity, is interpolated monthly data. The forcing data is supplied to the model on the
regular atmospheric grid used in the coupled setup during AWI-CM-1-1-MR's CMIP6 simulations
(Semmler et al., 2018) and interpolated to the multi-resolution grid used in the respective simulations by
FESOM. Runoff data is a monthly climatology and dynamic ice sheet coupling is not included, meaning
the freshwater influx from the Antarctic continent does not react to warming which may impact certain
processes, such as the timing and intensity of sea ice loss (Pauling et al., 2017; Bronselaer et al., 2018).
**2.3 Modeled ocean velocity data**

Geostrophic balance is an idealized approximation that does not match real ocean velocities for

several reasons, including the presence of ageostrophic flow, such as Ekman transport, as well as
assumptions made in the derivation of equations (1) and (2). Specifically, geostrophic balance between
the Coriolis effect and the pressure gradient is valid under the assumption that the curl of horizontal
velocities or vorticity is small relative to the magnitude of overall flow. In models, this assumption is
relatively close to reality in coarse-resolution simulations where geostrophic flow dominates, but on
higher-resolution meshes, where submesoscale flows are well-resolved, these omitted terms become
larger. Therefore, while using geostrophic velocities for both high-resolution and coarse-resolution
modeled datasets would be methodologically consistent, the error introduced would be systemically larger
for the finer-resolution dataset than the coarser. Therefore, we do not consider the use of geostrophic
velocities for both modeled datasets in this analysis to bring the data into closer agreement. Rather, for the
AWI-CM-1 dataset, where daily ocean velocities were not saved (Semmler et al., 2018), geostrophic
velocities derived from sea surface height with equations (1) and (2) are the best possible choice, and
fortunately, as described earlier, the error introduced by the assumptions of geostrophic balance will be
small. For the SO3 simulations, direct model output was saved and is preferred, particularly given the
high resolution of the mesh.
$u = -g/f * \partial SSH/\partial y$          (1)
$v = g/f * \partial SSH/\partial x$          (2)
The omission of Ekman transport, the primary source of ageostrophic oceanic flow from atmospheric
influences, can be relatively well addressed in the SO3 dataset by selecting modeled velocities just below
the Ekman layer. At depths of 25-30m below sea level, the bulk of Ekman transport can be avoided (Price
et al., 1987), while velocities should not substantially differ from those at the surface. What ageostrophic
flow remains in the model output velocities should be primarily large-scale and small relative to
geostrophic flow in the high-energy regions of the ocean, including the ACC (Yu et al., 2021).
**2.4 Altimetry data**

An observational data product of gridded, daily geostrophic velocities derived from along-track

satellite altimetry from crossover data is taken from the Data Unification and Altimeter Combination
System (DUACS) (Taburet et al., 2019). The gridded product has a resolution of 0.25 °, although
effective resolution at high latitudes may be much lower (Ballarotta et al., 2019). Recently, improved data
has become available in the ice-covered regions of the Southern Ocean (Auger et al., 2022), but does not
yet cover the full present-day simulated period (2016-2020) in this study. Absolute velocities from the
gridded altimetry product were used to calculate anomalies and EKE using equations (3) and (4) below
for consistency with the modeled dataset.
**2.5 EKE analysis**

Velocity anomalies are defined by subtracting the multi-year monthly climatology of each

respective 5-year period from daily velocities with equation (3).
$u'_i = u_i - \overline{u_m}$         (3)
Where $u_i$ is the daily zonal velocity, $'$ denotes an anomaly, and $\overline{u_m}$ is a monthly mean. For meridional
velocities (v) substitute u with v.
Eddy kinetic energy is calculated from ocean velocities according to equation (4).
$EKE_i = 0.5(u'^2_i + v'^2_i)$     (4)
Where ($_i$) denotes a daily value and ($'$) denotes an anomaly.
EKE was calculated on the native grid of each dataset and then interpolated to a 0.25 ° grid for all
analyses. In Fig. 1 and 3, EKE was first calculated on a daily timescale and coarsened to five-day means
before analysis to reduce computational costs during post-processing. Area-integrated EKE (Figure 1, 3)
is calculated by summing the area-weighted EKE of each grid cell in the study region defined as the zonal
band between 45 °S and 65 °S. The Brazil/Malvinas confluence region between 57 °E and 29 °E and
northward of 40 °S is removed to focus the study on a region with consistent physical drivers theorized to
be responsible for the changes in eddy activity (Beech et al., 2022). As a precaution, Each dataset was
linearly detrended before analysis in Fig. 1 and 3 to avoid artificially increasing the range of the later
distributions due to the accelerating climate change signal. Select statistical properties are reported in
Supplementary Tables 1-3 to indicate deviations from normality (D'Agostino and Belanger, 1990; Fisher,
1997) and autocorrelation (Durbin and Watson, 1950). Rather than attempt to manipulate the data to meet
certain statistical assumptions, complex statistical tests are avoided and the statistical properties reported
can be used to interpret the EKE data in a physical sense. EKE anomalies (Figure 1) were calculated by
subtracting the 2016-2020 mean of area-integrated EKE from the 5-day mean values of each period.
Normalized EKE was calculated by further dividing EKE anomaly by the standard deviation of EKE
during the 2016-2020 period. In Fig. 4, ensemble agreement is determined by ordering the $\Delta$EKE values
within each grid cell from lowest to highest, plotting the positive values in increasing order from left to
right and negative values in decreasing order from left to right.
**3   Results**
**3.1 Agreement with observations**

During the five-year period of overlap with observations, the SO3 simulation is a drastic

improvement on the AWI-CM-1 ensemble in reproducing median observed EKE (Figure 1a, c, note the
different y axes); only a slight underrepresentation of EKE remains in the SO3 simulation, although the
simulated distribution is somewhat distinct from observations. In comparison, the AWI-CM-1 ensemble,
being effectively eddy-permitting in the Southern Ocean, underrepresents observations by approximately
55% (Figure 1a, c, note the different y-axis). EKE in SO3 appears more variable than the observations
considering its larger range, (Figure 1c, e), and in general, the modeled datasets display greater deviations
from a normal distribution than the observations (Figure 1a, b, c; Supplementary Table 2). Nonetheless,
relative to the AWI-CM-1 model bias and the magnitude of EKE resolved, the ensemble spread within the
AWI-CM-1 dataset is small (Figure 3), suggesting that a single ensemble member of five years duration is
sufficient to assess how well a model captures the magnitude of overall Southern Ocean EKE (Figure 1c).

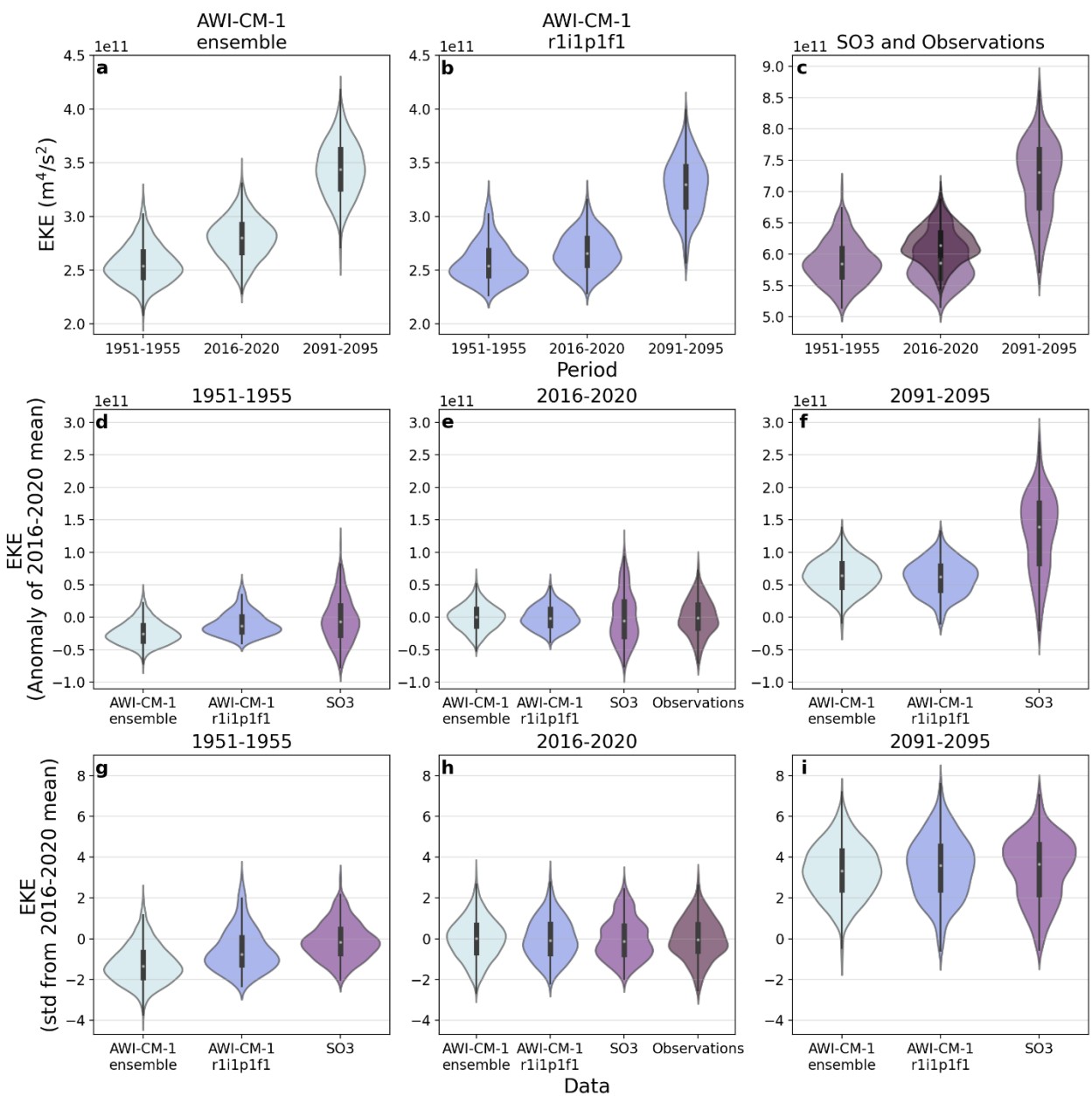

**Figure 1. Violin plots of area-integrated Southern Ocean EKE in simulations and observations.**
Central points of each plot indicate the median, thick bars span the first and third quartiles, thin bars span the range, and the violin body is a kernel density estimation of the data. **a-c)** Magnitudes of area-integrated EKE (note the different y axes) **a)** The AWI-CM-1 ensemble. **b)** the first member of the AWI-CM-1 ensemble, from which the SO3 simulations take their atmospheric forcing. **c)** The SO3 simulations and observations. **d-f)** Anomalies relative to the 2016-2020 mean of area-integrated EKE for each dataset

respectively. **d)** 1951-1955. **e)** 2016-2020. **f)** 2091-2095. **g-i)** Normalized values relative to the mean and
standard deviation of EKE during the 2016-2020 period for each dataset respectively. **g)** 1951-1955. **h)**
2016-2020. **i)** 2091-2095.

From a regional perspective, the SO3 simulation accurately reflects local magnitudes of observed

EKE and also generally captures the spatial distribution well (Figure 2). However, there are regional
shortcomings, such as between 90 and 145 °E. Grid resolution in this region should be sufficient to
resolve eddy activity (Supplementary Figure 1), indicating that the bias arises from another source. In the
AWI-CM-1 ensemble, the regional representation of EKE reinforces a broad underrepresentation relative
to observed magnitudes, but the major geographic features of eddy activity are fairly well represented
(Figure 2). Once again, the ensemble spread within the AWI-CM-1 simulations reveals remarkable
consistency, this time in terms of the spatial pattern and regional magnitudes (Supplementary Figure 2),
reinforcing the conclusion that a single ensemble member of five years duration is sufficient to assess the
mean state of EKE in the Southern Ocean. The consistency of the AWI-CM-1 ensemble further suggests
that regional shortcomings in eddy activity in the SO3 simulations are not a product of variability within a
single realization of Southern Ocean conditions (Supplementary Figure 2).

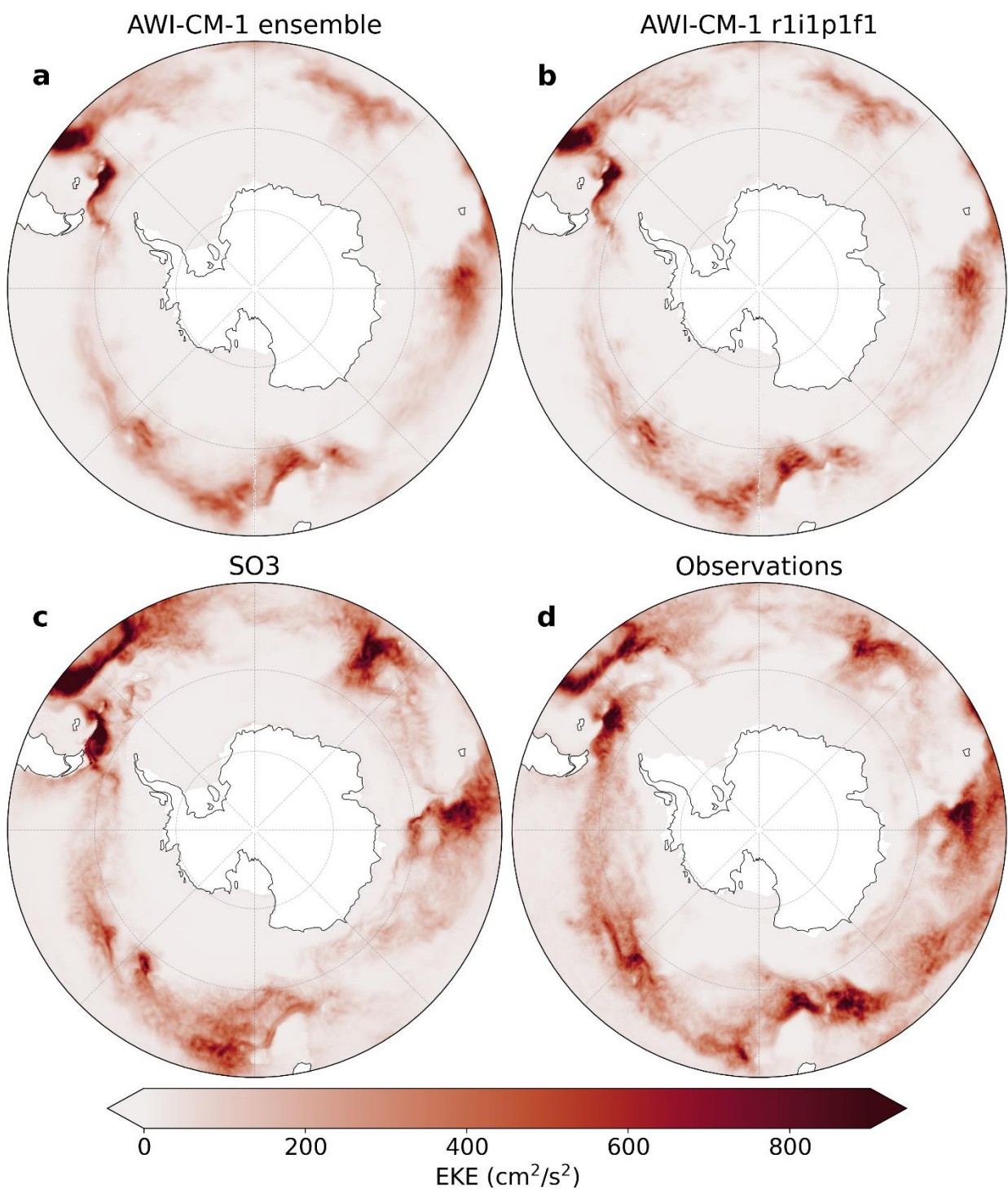


**Figure 2. Mean eddy kinetic energy between 2016 and 2020. a)** The AWI-CM-1 ensemble. **b)** The first

member of the AWI-CM-1 ensemble. **c)** The SO3 simulation. **d)** The gridded satellite altimetry dataset.


**3.2 EKE change and significance**

Southern Ocean eddy activity has been shown to intensify over the recent decades both using satellite altimetry (Martínez-Moreno et al., 2021), and the complete AWI-CM-1-1 dataset from CMIP6 (Beech et al., 2022). Even after reducing the AWI-CM-1 CMIP6 dataset to five-year periods preceding the apparent change (1951-1955) and at the end of the altimetry era (2016-2020), this intensification is still discernable within the AWI-CM-1 ensemble (Figure 1a). Despite this, the SO3 simulations do not demonstrate any substantial change in EKE magnitude over the same period (Figure 1). Further reducing the ensemble to its individual members (Figure 3), the EKE rise is still relatively robust in each case, including clear separation of the datasets considering the median, mode, and distribution of the data. However, the first ensemble member, from which the atmospheric forcing of SO3 is taken, demonstrates less EKE rise than the ensemble average (Figure 3), suggesting that natural variability in atmospheric conditions may contribute to the disagreement. Further investigation reveals several differences between the SO3 simulations and the AWI-CM-1 ensemble members that may play a role. Mean zonal ocean velocity in SO3 is faster and broader than the AWI-CM-1 ensemble (Supplementary Figure 3), meaning wind speed intensification may be misaligned with peak ocean velocities in SO3, particularly around 47 to 51 °S. Moreover, considerably less zonal wind stress is imparted to the ocean in SO3 despite identical wind speeds as the first AWI-CM-1 ensemble member (Supplementary Figure 4), possibly due to the higher ocean surface velocity.

The intensification of EKE becomes clear in both the AWI-CM-1 ensemble (Figure 1a), its members (Figure 3), and the SO3 simulations (Figure 1c) by the end of the 21$^{st}$ century. Over this period, the variability of EKE, indicated by the range of the distribution, also increases for each dataset (Figure 1f, i). EKE rise in SO3 is approximately twice that of the AWI-CM1 ensemble in absolute terms (Figure 1f), but expressing EKE as a relative value normalized by the mean and standard deviation of each dataset during the observational period (Figure 1g, h, i), reveals greater consistency between the changes until the end of the 21$^{st}$ century. EKE in each dataset appears to increase by approximately 3.5 standard deviations,

and the range of EKE distributions increases by approximately two to three standard deviations (Figure
1h, i). However, the datasets also tend to become more autocorrelated, which can inflate the distribution
range (Supplementary Tables 1, 3).

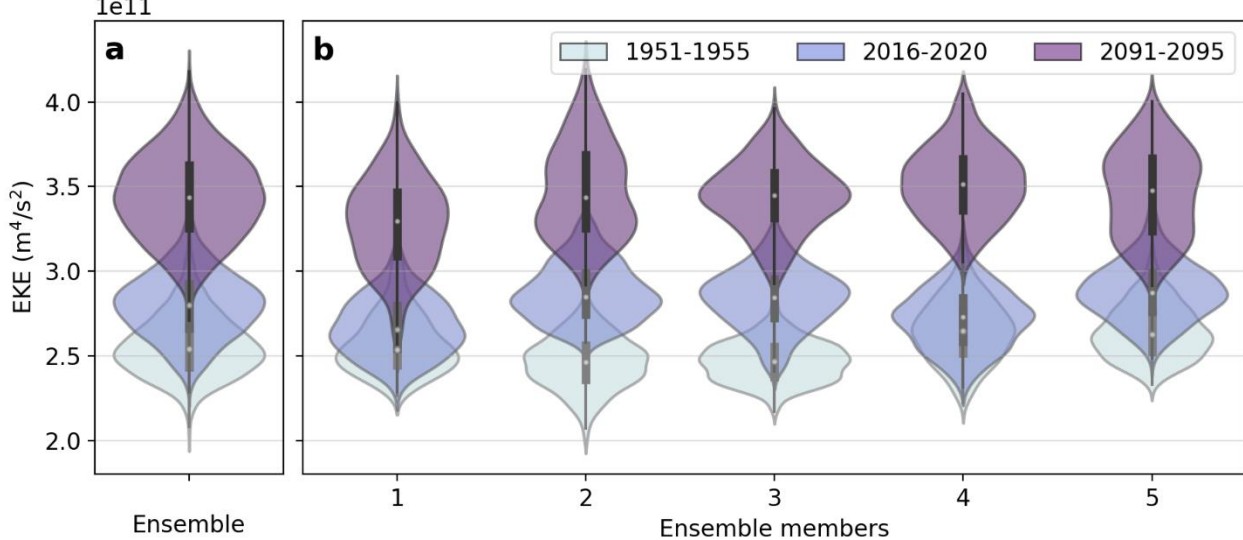


**Figure 3. Ensemble spread of EKE in AWI-CM-1. a)** Violin plots of area-integrated Southern Ocean
EKE in the AWI-CM-1 ensemble. **b)** Violin plots of mean Southern Ocean EKE in each member of the
AWI-CM-1 ensemble. Grey plots represent the period 1951-1955, blue plots represent 2016-2020, and
purple plots represent 2091-2095.
Before considering the regional impacts of warming on EKE in the SO3 simulations, it is useful
to refer to the ensemble spread within the AWI-CM-1 simulations to approximate the reliability of a
single ensemble member in revealing the ensemble-mean change as an analogue to the signal-to-noise
ratio. At 1 °C of warming, EKE change in the ensemble is weak, with at least one ensemble member
tending to show little or no EKE change in most regions (Figure 4a,c). Only a few clear patterns of
change emerge throughout the ensemble, namely the regions of EKE intensification downstream of the
Kerguelen Plateau and the Campbell Plateau where four to five out of five ensemble members show clear
EKE intensification (Figure 4a). It should be noted that even in these regions of relatively high confidence
(4 to 5 ensemble members, Figure 4a) EKE rise can be interspersed with lower-confidence (1 to 2
ensemble members, Figure 4c) EKE decline; this is also illustrated by the ensemble mean changes
themselves (Supplementary Figures 5, 6). Despite this, the consistency of EKE rise in these regions, and
their geographic positions in already EKE-rich regions, suggests that the intensification patterns are
robust changes within substantial noise. This level of noise suggests that EKE changes in the SO3
simulations at 1 °C of warming will be difficult to distinguish from natural variability when taken on their
own; indeed, in the SO3 simulations, the large variability of both sign and magnitude of change within
relatively small spatial scales does not lend confidence to any significant change at 1 °C of warming
(Figure 5c). However, building on the changes observed in the AWI-CM-1 ensemble, the intensification
of EKE downstream of the Kerguelen and Campbell Plateaus seems to be reinforced by the high-
resolution simulations.

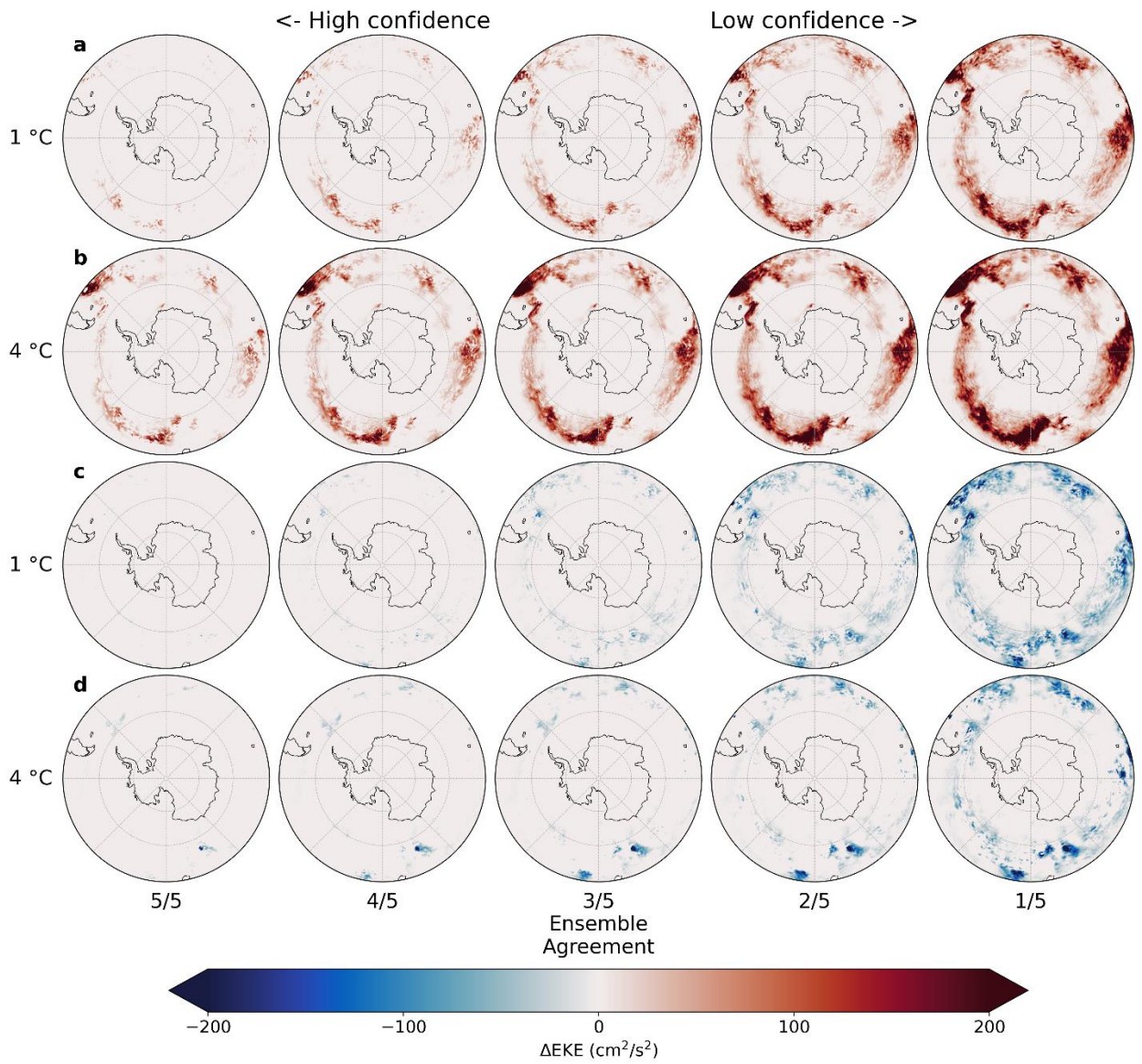

**Figure 4. Ensemble agreement regarding EKE change**. EKE rise (a, b) and decline (c, d) within the

AWI-CM-1 ensemble after one (a, c) and four (b, d) °C of warming or between 1951-1955 and 2016-

2020 and 2091-2095, respectively, arranged in order of decreasing ensemble agreement regarding change

in each grid cell. Ensemble agreement refers to the number of ensemble members that simulate at least the

pictured magnitude of mean EKE rise or decline for each grid cell. Mean EKE change is defined as the

difference of mean EKE between 1951-1955 and each of the two latter periods, as in Supplementary

Figures 5 and 6 but arranged in ascending order of magnitude for each grid cell and for positive and

negative signs separately. Rank 5/5 indicates the lowest magnitude of mean EKE rise (a, b) or decline (c,

d) within the ensemble for a given grid cell, meaning the entire ensemble agrees on at least this much
change. Rank 1/5 indicates the highest magnitude of EKE rise or decline within the ensemble for each
grid cell, representing the upper limit of projected EKE change.

At 4 °C of warming, change in eddy activity becomes clearer; EKE intensification downstream of

the Kerguelen and Campbell Plateaus is now consistent throughout the entire AWI-CM-1 ensemble, along
with additional intensifications south of the Falkland/Malvinas Plateau, around the Conrad Rise, and
along the Antarctic Slope Current at approximately 5 °E (Figure 4b). Four fifths of the ensemble also
include a broad increase in EKE throughout the ACC across most longitudes. Interestingly, a consistent
pattern of EKE decline also emerges upstream of the Campbell Plateau in the entire ensemble (Figure 4d).
The spatial pattern of EKE rise is relatively consistent regardless of confidence, with only the magnitude
increasing in the lower confidence composites (Figure 4b). The same tendency is observable between the
EKE changes at 1 and 4 °C of warming, where the magnitude of change is greater after further warming
but follows the same spatial pattern. Thus, regions of intensification can be identified more reliably than
the magnitude of change and tend to be concentrated where flow interacts with topographic features, in
already eddy-rich regions (Figure 2). Conversely, low confidence EKE decline appears nearly throughout
the Southern Ocean in at least one ensemble member, but only consistently upstream of the Campbell
Plateau and, to a far lesser extent, downstream of the Drake Passage and Campbell Plateau (Figure 4d).
Changes of negative sign tend to be of lower magnitude at 4 °C of warming than at 1 °C. This suggests
that the general EKE response to climate change in the Southern Ocean is that of intensification, and the
interspersed signals of decline tend to be the result of natural variability. Yet, small regions of high-
confidence EKE decline also appear. Consequently, it would be difficult to confidently separate reliable
EKE change from natural variability in simulations without an ensemble to compare with. In the SO3
simulations, EKE rise downstream of the Drake Passage and Kerguelen and Campbell Plateaus is
substantial (Figure 5f). EKE rise is also projected south of the Falkland/Malvinas Plateau, around the
Conrad Rise, and along the Antarctic Slope Current at approximately 5 °E, and a slight EKE decline
appears upstream of the Campbell Plateau. All of this is comparable to the AWI-CM-1 ensemble, and the
interspersed areas of EKE decline within these regions, for example, around the Conrad Rise, are not
improbable based on the example set by AWI-CM-1 (Figure 4d). However, considering that some high-
confidence EKE decline is present in the AWI-CM-1 ensemble, it is difficult to confidently dismiss
regional EKE decline in the SO3 simulations as noise.

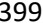


**Figure 5. EKE change.** Spatial representations of the difference in EKE between **(a-c)** 1951-1955 and
2016-2020, **(d-f)** 1951-1955 and 2091-2095. **a,d)** The AWI-CM-1 ensemble. **b,e)** the first member of the
AWI-CM-1 ensemble. **c,f)** The SO3 simulations.

**4 Discussion**

Intensification of eddy activity in the Southern Ocean is now widely accepted as a consequence of anthropogenic climate change (Hogg et al., 2015; Patara et al., 2016; Martínez-Moreno et al., 2021; Beech et al., 2022), and is understood to be caused primarily by stronger westerly winds imparting more energy to the Antarctic Circumpolar Current (Munday et al., 2013; Marshall, 2003). The results presented here reinforce the notion of EKE intensification and further project increased EKE variability as the climate warms (Figure 1, 3). By expressing EKE change in terms of ensemble agreement on a cell-by-cell basis, the results presented here are also able to identify regions of reliable and substantial change as those where flow interacts with major bathymetric features and high eddy activity is already known to occur (Figure 4). Analysis of regional changes within the Southern Ocean eddy field has generally been limited to regions defined by oceanic sectors (Atlantic, Indian, Pacific) (Hogg et al., 2015), or incremental longitudinal delimitations (Patara et al., 2016). In future research, regional analyses of the significance, rate, or cause of EKE trends could focus on the bathymetrically defined regions identified in this analysis to produce physically related and consistent results.

The consistency of the AWI-CM-1 ensemble in projecting clear EKE rise in the Southern Ocean as a whole suggests that a single ensemble member of five-years simulation length should be sufficient to reliably identify change, even after 1 °C of temperature rise. Despite this, the SO3 simulations fail to reproduce the EKE rise that is already observable through observations (Martínez-Moreno et al., 2021). A potential source for this discrepancy is the uncoupled model setup in the SO3 simulations which omits ocean-atmosphere feedbacks. In this regard, the SO3 simulations experience lower wind stress imparted to the ocean surface than AWI-CM-1 ensemble member one by the same surface winds (Supplementary Figure 4), and a mismatch between peak zonal wind speeds and mean zonal ocean velocities (Supplementary Figure 3). Confounding the comparison further, is the fact that strengthening winds can both increase and dampen eddy activity; as westerlies intensify, the additional energy imparted to the ocean is expected to strengthen eddy activity (Munday et al., 2013; Meredith and Hogg, 2006), but winds

are also known to dampen mesoscale activity through eddy killing (Rai et al., 2021) and this impact is
greater in uncoupled model configurations (Renault et al., 2016). While the lack of change at 1 °C is
difficult to explain, the disagreement is limited to these more subtle changes and the simulations tend to
agree on the strong EKE rise at 4 °C of warming.

The remaining discrepancies between eddy activity in SO3 and observations are relatively small,

but exploring potential sources of disagreement may help to interpret the simulations and guide future
modeling endeavors. Greater skew in the distribution of EKE in the modeled dataset (Supplementary
Table 2) could reflect multiple modes of circulation or seasonality. While seasonality of eddy activity in
the ACC is low, seasonal ice cover likely affects eddy activity in the modeled dataset, and certainly
affects the observational dataset by producing gaps in its spatio-temporal coverage. Beyond differences in
skew, this could contribute to the greater range of EKE seen in the SO3 simulations by systemically
obscuring seasonal conditions from the observational dataset. Regional deficiencies of EKE in SO3 could
be explained in terms of grid resolution outside of the study region; resolving the first Rossby radius of
deformation with at least two grid points is not enough to comprehensively reproduce mesoscale activity
(Hallberg, 2013; Sein et al., 2017), and grid refinement may need to be expanded to upstream regions that
impact eddy dynamics in the Southern Ocean. Other sources of bias may include ocean-atmosphere
interactions which are absent or unrealistic within the uncoupled simulations (Byrne et al., 2016; Rai et
al., 2021; Renault et al., 2016). As well, some small-scale, slow-to-equilibrate ocean processes may be
resolved in the high-resolution simulations, but not be integrated long enough for their effects to impact
eddy activity (van Westen and Dijkstra, 2021; Rackow et al., 2022). Finally, the gridded altimetry product
itself may be responsible for some disagreement, as the along-track data is known to underrepresent eddy
activity at scales less than 150km and 10 days (Chassignet and Xu, 2017), which will be particularly
impactful at high latitudes.

To distinguish a meaningful signal of anthropogenic impacts from natural variability, this

analysis relies primarily on consistency among ensemble members (Figures 3, 4). This is distinct from
more traditional methods like assessment of error relative to observations or ensemble mean, commonly
applied to weather forecasting (Ferro et al., 2012), but can be compared to measures of ensemble
agreement used extensively in the IPCC reports (Fox-Kemper et al., 2021). Performance evaluation
relative to observations would undoubtedly point to the high-resolution simulation as superior due to the
drastic underrepresentation of EKE in the eddy-permitting ensemble (Figure 1). Yet, the effects of climate
change are still apparent in the AWI-CM-1 ensemble (Figure 1, 5), and the AWI-CM-1 dataset has been
used to make similar projections of EKE already (Beech et al., 2022). Moreover, the eddy response to
forcing seems to be consistent between the model resolutions when expressed in relative (Figure 1g, h, i),
rather than absolute terms (Figure 1a, b, c). While more verification of this result is necessary both
regionally, and with other models, these results suggest that eddy-permitting resolutions can be
interpreted with their shortcomings in mind in order to discern the real-world implications: as is often
necessary with model data. Thus, based on the test case of the Southern Ocean, the usefulness of the
AWI-CM-1 ensemble and the effectiveness of model simulations in identifying physically significant and
reproduceable impacts of climate change may be greater than would be identified using traditional
methods and comes at a much lower cost relative to the eddy-resolving simulations.

This study has focused on EKE as an evaluation metric for the simulations since mesoscale

activity is the primary motivation for increasing ocean model resolution. It has stopped short of assessing
the improvements that resolving the mesoscale has on climate and ocean dynamics, many of which are
discussed in detail elsewhere (eg. Hewitt et al., 2017). Rather than repeat an assessment of the benefits of
resolving smaller scales, we assume that the accurate reproduction and evolution of eddy activity
indicates that these improvements are transferred to broader processes. Certainly, inaccurate simulation of
the mesoscale would raise questions regarding the improvements that this mesoscale activity should have
on the simulations as a whole. Nonetheless, further evaluation of the modeling approaches employed in
this study will be necessary to determine if these methods are appropriate for studying broader elements
of the climate system. Since the high-resolution simulations derive their deep-ocean climate primarily
from the medium-resolution spin-up simulation, improving the initialization process (Thiria et al., 2023)
may be the critical barrier to extending these results from the mixed layer to the deeper ocean.
**5 Conclusion**

Resolving the ocean mesoscale has become a focus for the climate and ocean modeling

community as computational capabilities expand and models become increasingly complex. The benefits
that explicitly resolved eddy activity can have on climate simulations are clear (Hewitt et al., 2017; Sein
et al., 2017) along with the impact that mesoscale variability has on local (Lachkar et al., 2009; Wang et
al., 2017) and global environments (Falkowski et al., 1991; Sallée et al., 2012). However, state-of-the-art
climate models will be unable to fully resolve the mesoscale for the foreseeable future, particularly in
large-scale modeling endeavors such as CMIP (Hewitt et al., 2020). Thus, modelers must make informed
choices regarding the explicit processes needed to answer research questions and where resources must be
allocated to achieve specific goals. Existing analysis of resource allocation has typically addressed short-
term weather forecasting or the ability to reproduce observations with low error (Ferro et al., 2012), but
the question of how to best allocate resources for climate change impact assessment remains. This study
has applied several cost-efficient modeling approaches to an analysis of the impacts of climate change on
a key focus of high-resolution modeling: the mesoscale. Applying these results to broader climate change
impact studies should improve the efficiency of resource allocation and focus modeling studies.
Resolution can be dynamically adjusted both spatially, by focusing resources in study regions and where
they are necessary to resolve local dynamics, and temporally, by allowing lower-resolution workhorse
configurations to perform spin-up and transient runs. Limited simulation length and ensemble size can be
sufficient for certain research questions and validation, but simulations must ultimately be designed to
meet their specific goals. Where resources are limited, studies may best include a combination of eddy-
resolving simulations able to fully capture the local eddy field, as well as eddy-permitting simulations that
can attest to the significance of results through consistency and repetition.

503   This work represents a contribution to the growing wealth of research that points to an

504 intensification of eddy activity in the Southern Ocean (Hogg et al., 2015; Martínez-Moreno et al., 2021;

505 Beech et al., 2022). The further conclusions that EKE variability may increase and that EKE

506 intensification appears concentrated in key regions based on topography can both expand the present state

507 of knowledge, as well as direct future research. The cost-efficient modeling approaches of regional grid

508 refinement, reduced-resolution spin-up and transient runs, and limited simulation lengths distinguished by

509 longer periods of change are demonstrated to be effective at reproducing change within a more traditional

510 eddy-permitting ensemble. When resources are limited and resolution demands are high, these approaches

511 can be adapted to address specific research questions. Where assessing the robustness of change is

512 critical, the complimentary eddy-permitting ensemble represents an effective, low-cost supplement to the

513 high-resolution simulations.

**Data Availability**

Geostrophic velocities derived from satellite altimetry data are publicly available at https://doi.org/10.48670/moi-00148. Daily sea surface height data from AWI-CM-1-1-MR in CMIP6 used to compute geostrophic velocities in this study is archived at the World Data Center for Climate at the DKRZ (https://doi.org/10.26050/WDCC/C6sCMAWAWM, https://doi.org/10.26050/WDCC/C6sSPAWAWM) (Semmler et al., 2022a, b). Model output from AWI-CM-1-1-MR in the CMIP6 framework, including all variables used to force the standalone ocean simulations conducted for this study, is publicly available at https://doi.org/10.22033/ESGF/CMIP6.359 (Semmler et al., 2018). Eddy kinetic energy datasets calculated from FESOM output velocities are available at https://doi.org/10.5281/zenodo.8046792 (Beech, 2023b).

**Code Availability**

Source code for the ocean model FESOM2 is available at (https:/doi.org/ 10.5281/zenodo.7737061) (patrickscholz et al., 2023). Code used for data analysis and visualization in this study is publicly available at (https://doi.org/10.5281/zenodo.8046782) (Beech, 2023a). Code used to calculate geostrophic velocities from sea surface height data from AWI-CM-1-1-MR is available from https://doi.org/10.5281/zenodo.7050573.

**Author Contributions**

NB, TJ, TR, and TS conceived of the study. NB carried out the simulations, analyzed the data, and drafted the manuscript. All authors reviewed the manuscript.

**Competing Interests**

The authors declare no competing interests.

**Acknowledgements**

The work described in this paper has received funding from the Helmholtz Association through

the project 'Advanced Earth System Model Capacity' (project leader: T.J., support code: ZT-0003) in the
frame of the initiative 'Zukunftsthemen'. The content of the paper is the sole responsibility of the authors
and it does not represent the opinion of the Helmholtz Association, and the Helmholtz Association is not
responsible for any use that might be made of information contained. TJ acknowledges the EERIE project
funded under the EU Horizon Europe programme (grant number 101081383). TR acknowledges support
from the European Commission's Horizon 2020 collaborative project NextGEMS (grant number
101003470). This work used resources of the Deutsches Klimarechenzentrum (DKRZ) granted by its
Scientific Steering Committee (WLA) under project ID 995. The CMIP data used in this study were
replicated and made available by the DKRZ.

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
