# Peer review of "Exploring the ocean mesoscale at reduced computational cost with FESOM 2.5: efficient"

_EGUsphere, 2023_

## Referee Comment (RC1)

Review of "Resolving the mesoscale at reduced computational cost with FESOM 2.5: efficient modeling approaches applied to the Southern Ocean" by Nathan Beech et al.

This paper explores the eddy activity in a variable-resolution Southern Ocean configuration with up to 3km gridcells (SO3), using FESOM2.5, using atmospheric forcing data from three different five-year periods. The comparison data set of simulations is AWI-CM-1 medium-resolution configurations with nominally ¼ degree (~25km) gridcells. Three central questions are explored: (1) How does the SO3 configuration alter eddy activity over medium resolution; (2) How do both of those compare to satellite altimetry for the present-day simulations; and (3) How do the medium-resolution and SO3 change with climate change.

A major challenge of running at 3km resolution is the computational requirement. This experimental design makes the best use of available compute time by studying the eddy kinetic energy, which adjusts quickly, rather than ocean climatology which may take over 100 years to adjust (ocean water mass temperature and salinity, mixed layer depth, frontal location). The authors clearly explain this strategy, and I think they make the best possible use of 18 years of high-resolution simulations for a scientific study. Other modeling centers also struggle with the computational cost of long spin-up times at high resolution. The authors downscale their initial conditions from lower resolution FESOM runs for each period ("a semi-cold start-up") to assist the spin-up process, which is only one year.

This paper very well written. The introduction and methods provide sufficient background, detail, and references. The analysis can be improved with the two major points below, but the included plots are well organized and nicely labelled. The English writing is excellent. This paper will be of interest to GMD readers, and ocean modelers more generally. I am happy to see a publication with scientific results from variable-resolution ocean models, down to 3km resolution!

Major Comments
1. In my view, the big missing piece of analysis for this study is the magnitude of the westerlies in the atmospheric forcing data. Figure 1 is a wonderful, clear summary of the model behavior, and shows that eddy activity increases substantially in the 2090s simulations for all resolutions and shows higher EKE at high resolution. The presumable cause of the gains from present day to 2090s is stronger westerly winds, as referenced in the Munday et al 2013 and Marshall 2003 papers. Please add a figure with violin plots or similar analysis of the distribution of Southern Ocean westerly winds used in the model atmospheric forcing data. This could look like Fig 1 panel c for the different time simulations, plus winds from AWI-CM-1 simulations, which had an active atmosphere.
2. There is no mention of the sea ice, other than that it is included in the model. Given that the sea ice covers part of the Southern Ocean every winter, presumably it has some modulating effect on the eddy kinetic energy. For example, "Generally, EKE is stronger when sea ice concentration is low versus times of dense ice cover." and "Consolidated sea ice dampens eddy kinetic energy by reducing the atmosphere-ocean momentum transfer that drives part of the mesoscale variability, for example, along Arctic shelf

breaks" in Wilken-Jon von Appen et al. 2022, although that was specifically an Arctic study. So a second hypothesis is that sea ice cover is substantially reduced in 2091, leading to higher EKE. Which is it, increased winds or reduced sea ice? It may be beyond the scope of this paper to nail that down completely, but I think it is worth a literature search and discussion on this point. Like point 1, you could include the winter and summer Southern Hemisphere sea ice area (and perhaps volume) from the different simulations. I'm sure the sea ice for the 2090s simulations is greatly reduced. Then you have two potential culprits for the increased EKE. They probably work together – in the 2090s there is (presumably) both stronger winds *and* a more direct influence by the winds due to less sea ice.

Minor comments
L14 "eddy-present" is a new adjective for me. Is that standard? I've heard "eddy permitting" for 1/10 degree, but not "eddy-present" for ¼ degree. I see it appears in Moreton et al. 2020 for ¼ degree, so I must just be behind the times.

L179 Please add the range of core counts you typically run on, and the core count for the 0.65 SYPD.

L361 Is -> is (the only grammatical or spelling mistake I found in the whole paper!)

---

## Author Comment (AC1)

**Response to Reviewer #1**

Many thanks for your positive reception and encouraging review of our manuscript.

Review of "Resolving the mesoscale at reduced computational cost with FESOM 2.5: efficient modeling approaches applied to the Southern Ocean" by Nathan Beech et al.

This paper explores the eddy activity in a variable-resolution Southern Ocean configuration with up to 3km gridcells (SO3), using FESOM2.5, using atmospheric forcing data from three different five-year periods. The comparison data set of simulations is AWI-CM-1 medium-resolution configurations with nominally ¼ degree (~25km) gridcells. Three central questions are explored: (1) How does the SO3 configuration alter eddy activity over medium resolution; (2) How do both of those compare to satellite altimetry for the present-day simulations; and (3) How do the medium-resolution and SO3 change with climate change.

A major challenge of running at 3km resolution is the computational requirement. This experimental design makes the best use of available compute time by studying the eddy kinetic energy, which adjusts quickly, rather than ocean climatology which may take over 100 years to adjust (ocean water mass temperature and salinity, mixed layer depth, frontal location). The authors clearly explain this strategy, and I think they make the best possible use of 18 years of high-resolution simulations for a scientific study. Other modeling centers also struggle with the computational cost of long spin-up times at high resolution. The authors downscale their initial conditions from lower resolution FESOM runs for each period ("a semi-cold start-up") to assist the spin-up process, which is only one year.

This paper very well written. The introduction and methods provide sufficient background, detail, and references. The analysis can be improved with the two major points below, but the included plots are well organized and nicely labelled. The English writing is excellent. This paper will be of interest to GMD readers, and ocean modelers more generally. I am happy to see a publication with scientific results from variable-resolution ocean models, down to 3km resolution!

Major Comments

1. In my view, the big missing piece of analysis for this study is the magnitude of the westerlies in the atmospheric forcing data. Figure 1 is a wonderful, clear summary of the model behavior, and shows that eddy activity increases substantially in the 2090s simulations for all resolutions and shows higher EKE at high resolution. The presumable cause of the gains from present day to 2090s is stronger westerly winds, as referenced in the Munday et al 2013 and Marshall 2003 papers. Please add a figure with violin plots or similar analysis of the distribution of Southern Ocean westerly winds used in the model atmospheric forcing data. This could look like Fig 1 panel c for the different time simulations, plus winds from AWI-CM-1 simulations, which had an active atmosphere.

We agree that the particular behaviour of the westerlies is important information that was missing from the original manuscript. We have added a figure that includes the changes in mean zonal wind speed between 1951-1955 and the two later periods for each CMIP6 ensemble members and the ensemble mean. The wind forcing for the SO3 simulations is taken from the first ensemble member of the AWI-CM-1 ensemble. We also include plots of the mean zonal wind stress to the ocean in SO3 and the AWI-CM-1 ensemble, which can be different despite the identical wind forcing.

2. There is no mention of the sea ice, other than that it is included in the model. Given that the sea ice covers part of the Southern Ocean every winter, presumably it has some modulating effect on the eddy kinetic energy. For example, "Generally, EKE is stronger

when sea ice concentration is low versus times of dense ice cover." and "Consolidated sea ice dampens eddy kinetic energy by reducing the atmosphere-ocean momentum transfer that drives part of the mesoscale variability, for example, along Arctic shelf breaks" in Wilken-Jon von Appen et al. 2022, although that was specifically an Arctic study. So a second hypothesis is that sea ice cover is substantially reduced in 2091, leading to higher EKE. Which is it, increased winds or reduced sea ice? It may be beyond the scope of this paper to nail that down completely, but I think it is worth a literature search and discussion on this point. Like point 1, you could include the winter and summer Southern Hemisphere sea ice area (and perhaps volume) from the different simulations. I'm sure the sea ice for the 2090s simulations is greatly reduced. Then you have two potential culprits for the increased EKE. They probably work together – in the 2090s there is (presumably) both stronger winds *and* a more direct influence by the winds due to less sea ice.

It is a good point that sea ice decline is almost certainly playing a role in the intensification of eddy activity in the seasonal sea-ice zone around the Antarctic. However, the area affected by sea-ice decline contains very little eddy activity relative to the highly-energetic ACC which is already almost entirely sea-ice free. We also limit our study region to areas north of 60S which will again reduce the effects of declining sea ice. Therefore, the changes we identify in this assessment will be dominated by the effects of wind stress. However, work is already underway assessing higher-latitude changes in this dataset. Nonetheless, we have added a discussion point on how seasonal ice cover affects EKE, particularly in the observational dataset through data availability. This is at line 432.

Minor comments

L14 "eddy-present" is a new adjective for me. Is that standard? I've heard "eddy permitting" for 1/10 degree, but not "eddy-present" for ¼ degree. I see it appears in Moreton et al. 2020 for ¼ degree, so I must just be behind the times.

"Eddy-present" and "eddy-rich" are preferred by some to the roughly equivalent "eddy-permitting" and "eddy-resolving". None of these terms are necessarily standard, which may be a weakness in the modelling community, but there is no standard alternative as far as we are aware. Generally, eddy-present refers to a partially-resolved eddy field and eddy-rich refers to a well-resolved eddy field. The exact grid resolution that this corresponds to can vary based on the study region, so it is possible that you have seen the same terms used for different model resolutions. What is relevant is how they can be used to describe model resolutions in terms of the ability to simulate the mesoscale. (Hallberg, 2013) and (Sein et al., 2017) are two good investigations of this topic.

L179 Please add the range of core counts you typically run on, and the core count for the 0.65 SYPD.

The core count we ran on is at line 180 - 8192. This is the only setup we have used.

L361 Is -> is (the only grammatical or spelling mistake I found in the whole paper!)

The capitalization has been corrected.

Response to reviewer #2

We appreciate your thorough review of our paper and constructive comments, particularly regarding the clarity of our methods and the justification of some decisions. Your thoughts have identified several areas for improvement.

This is a fairly straight forward and interesting paper that makes a useful contribution to understanding how constrained computational resources should be used in climate change scenarios. The paper focuses upon Eddy Kinetic Energy (EKE) and how to make reliable estimates of how this might change as a result of climate change. To do so, the authors use an ensemble of eddy-permitting models alongside a configuration specifically designed to better represent the Southern Ocean. I found the conclusion that only a single realisation of a given model is needed to appreciate how climate change affects EKE compelling. This paper makes a useful contribution to on-going discussion in the literature regarding the impact of climate change on the Southern Ocean and its mesoscale eddy field.

Most of my comments regarding the paper are quite minor. There isn't complicated analysis here, which is a strength of the paper as it makes it simple to understand. However, the paper's clarity could be improved, particularly in Section 2. In addition, there are some choices in the analysis, such as using geostrophic surface EKE for one model and subsurface full EKE for another, that need stronger justification.

To address these points, please see our responses to the specific comments below.

As a final point, from the title I was expecting something more technical, like using single precision maths, alongside the use of FESOM's non-uniform resolution. The paper is really using a carefully designed spinup procedure with an equally carefully put together finite element model and forcing dataset. The title should be altered to reflect this.

We have changed the title to reflect that the novel points of the simulations mainly relate to the procedure.

**Specific Comments**

In Section 2.1 it is currently tricky to understand which model is which resolution and whether they are coupled or ocean-only. After reading the paper there should be three; the high resolution SO3 configuration, a medium-resolution ocean-only model used to supply SO3 with initial conditions, and the coupled AWI-CM-1 ensemble. At the moment this isn't entirely clear up front. The first paragraph made me think that SO3 was coupled, while the third paragraph makes it clear that it is ocean-only. It also isn't clear here that AWI-CM-1 is an ensemble instead of a single model run. In the case of AWI-CM-1, whilst details are elsewhere and a citation is given, it would be much clearer to tell the reader that this is a five member ensemble and how that ensemble is generated in a sentence or two. The "medium-resolution eddy-permitting, ocean-only transient simulation" isn't used in detail, but its existence is a little obfuscated here. At the beginning of the second paragraph it is stated that "model experiments with SO3 consist of a medium-resolution…", whilst the third paragraph tells us the SO3 is "the higher-resolution ocean grid". This can be easily remedied, but certainly caused me to pause on first read through.

Yes, there are three configurations as you have described and Section 2.1 has been substantially revised to clarify this. The first paragraph now states specifically that the simulations on the SO3

mesh are ocean-only and that the AWI-CM-1 dataset used is a five-member ensemble. Descriptions of the eddy-permitting transient simulation and the high-resolution simulations with SO3 have now been separated into the second and third paragraphs for clarity.

Section 2.4 describes how the geostrophic velocities are calculated to facilitate how the models are compared. There are a few issues here that should be dealt with. Firstly, no explanation is given why geostrophic surface velocity wasn't used for SO3. This would be a simpler way to guarantee that all ageostrophic motions are removed and make for a better comparison with the AWI-CM-1 ensemble and altimetry. If SSH is available for SO3, a quick comparison would validate the use of subsurface velocities instead. If there is a substantial difference, then this would require further work to ensure that the rest of the paper still stands. Secondly, equations (1) and (2) should be relocated to the middle of the first paragraph, where they are referred to. They are currently orphaned at the end of the paragraph. They are also incorrect; the pre-multiplier to the differential should be g/f, not gf, although I suspect this is purely a typo given that this would grossly alter the order of magnitude of the EKE.

We see that the use of geostrophic velocities for observations and the CMIP6 data and direct model output velocities for the SO3 data was not sufficiently explained in the original manuscript. We have added more details to the revised draft in section 2.3 and will include them here.

It is well understood that geostrophic balance is an idealized approximation that does not match real ocean velocities for several reasons, including the presence of ageostrophic flow, such as Ekman transport, as well as assumptions made in the derivation of equations 1 and 2. Specifically, geostrophic balance between the Coriolis effect and the force of gravity is valid under the assumption that the curl of horizontal velocities or vorticity is small relative to the magnitude of overall flow. In models, this assumption is relatively close to reality in coarse-resolution simulations where geostrophic flow dominates, but on higher resolution meshes where mesoscale and submesoscale movements are resolved, these omitted terms become larger. Therefore, while using geostrophic velocities for both high-resolution and coarse-resolution modeled datasets would introduce the same assumptions regarding geostrophic balance, the error introduced would be systemically larger for the finer-resolution dataset than the coarser. Therefore, we do not consider the use of geostrophic velocities for both modeled datasets in this

analysis to bring the data into closer agreement. Rather, for the CMIP6 dataset, where daily ocean velocities were not saved, geostrophic velocities are the best possible choice of data, and fortunately, as described earlier, the error introduced by the assumptions of geostrophic balance will be small. For the SO3 simulations, direct model output was saved and is preferred, particularly given the high-resolution of the mesh. Perhaps the weakest representation of 'reality' in this analysis, is the observational dataset, in which the error introduced by the aforementioned assumptions may be large, but no comparably comprehensive dataset for ocean velocities exists.

The method used to test ensemble agreement seems idiosyncratic. It isn't a standard procedure that I'm aware of, although this obviously doesn't guarantee that this isn't the case. I expected some sort of statistical analysis to confirm whether or not the ensemble members were in agreement and this, much simpler, approach took me by surprise. A more in-depth explanation of how to interpret what is being calculated might help. For example, should Figure 4 be interpreted as showing where the ensemble members share a common standard deviation? Does this measure imply anything about the range of values found as five days means?

Figure 4 can be compared to methods used throughout the IPCC reports to convey ensemble agreement for climate change projections (see Chapter 9 of the Sixth Assessment Report Figures 9.1-9.13). Where the authors of AR6 tend to use hatching over areas with less than 80% model agreement regarding the sign of change, we are able to display the exact proportion of ensemble members in agreement due to our much-reduced dataset. We also separate positive and negative trends to display regions of conflicting change (both positive and negative change in some ensemble members) which in the IPCC reports is sometimes indicated by cross-hatching. We have added this to the discussion at line 443.

We have added details to the figure caption of Figure 4 in order to clarify the points you mention. EKE rise and decline is calculated as the difference of mean EKE during the three periods entirely. Mean EKE is calculated from daily values, but this will not change the mean value of EKE throughout the entire period compared to the mean of 5-day mean values. The standard deviation of EKE is not considered in this figure.

In Section 4 it is hypothesised that the lack of EKE change in SO3 at 1oC could be due to a stronger wind stress causing an increase in relative wind stress damping. It is certainly the case that relative wind damping has an influence over Southern Ocean EKE (Munday et al., 2021). It

is a bit of a shame to leave this has a hypothesis; changes in relative wind damping can be tested by calculating the actual damping using Reynolds averaging of the geostrophic wind power input. In addition, the expected power budget of the Southern Ocean is wind power input being balanced by bottom kinetic energy dissipation due to friction (Abernathey et al., 2021). As such, it could also be the case that the alignment of SO3's wind stress and surface current is such that there isn't the increased power input required to drive a stronger eddy field.

It would be more satisfying to directly test whether the lower-than-expected EKE rise in the SO3 simulations at 1C is due to the standalone ocean causing more eddy-killing. However, the simulations we performed were not structured with this sort of test in mind, making it difficult to envision a reliable analysis. If we calculated changes in relative wind damping in the uncoupled SO3 simulations, we would still lack a comparable coupled simulation from which to assess the difference. The AWI-CM-1 ensemble that we primarily compare SO3 to would not be appropriate due to the drastically different grid resolution. Since Renault et al. (2016) have already demonstrated that an uncoupled simulation can expect more eddy-killing than a coupled simulation (27% in their study region), the contribution that we could potentially make would be to assess whether this value changes as relative wind stress changes. For this, a repetition of the SO3 simulations with a coupled atmosphere included would be a suitable comparison for our data, but it seems to me that this hypothesis could be tested at much lower computational cost using an idealized model configuration and varying surface winds. Thus, we leave this analysis for future research.

Your point about the alignment of the wind stress and surface currents is a good one and could potentially be the reason behind the lack of change in the SO3 simulations. We have added a figure to the supplementary information that shows this and included it in the text at line 314.

**Technical Corrections**

Line 12 : it would be less mysterious to tell us what the "cost-efficient, high-resolution modelling approaches" are at this point in the abstract.

We have modified the first sentence of the abstract to summarize the efficiency-maximizing strategies directly.

Line 18 : it is fairly normal to get about a 50% at 1/4o grid spacing. Perhaps mention that here?

In the interest of keeping the abstract as concise as possible, we have added this to the text at line 266.

Line 22 :"despite full ensemble agreement" regarding the project EKE increase?

Yes, this has been edited for clarity.

Line 47-50 : a very long sentence that could be split at "but shortcomings remain".

The sentence has been split.

Line 51 : "they" is ambiguous; what must vary in space?

This has been corrected to specify the model resolution.

Line 64-66 : mesoscale eddies also play an active role in setting the general stratification/pycnocline depth (see Marshall et al., 2002).

This point has been added.

Line 70-73 : a good point, not something I've seen phrased this way before.

Thank you.

Line 77-78 : I found this sentence ambiguous, it could be interpreted as meaning that the wind stress and heat fluxes, etc, won't change in a forced ocean simulation. In practise, they of course will, and this isn't what the authors mean. Its the prescribed atmospheric temperature, 10 m winds, etc, that can't change.

We have specified absolute surface winds will react to mesoscale activity in a coupled model.

Line 130, 140, 45 : why AWI-CM-1-1MR instead of AWI-CM-1?

In CMIP6, the naming conventions for contributions from AWI-CM and AWI-ESM included the version information (1-1) and the mesh (MR). There were other meshes included in the various contributions, and AWI-CM-1-1-MR produced more simulations (eg. different emissions scenarios, the preindustrial control, etc.) than those considered in this manuscript and named as AWI-CM-1 for brevity at line 119. When we refer to the entire suite of simulations produced by AWI-CM-1-1-MR rather than the specific set we analyze, we use the full name of the model. This is more appropriate at the lines you have pointed out because the paper being referenced includes details of the entire suite of simulations.

We recognize that this was not originally clear in the manuscript and we have specified that the simulations we refer to as AWI-CM-1 are a subset of AWI-CM-1-1-MR's CMIP6 contribution at the beginning of the subsection.

Line 135 : "the medium resolution: opening the sentence is repetitive, maybe replace it with SO3 to make it clear which model we're talking about.

This sentence refers to the transient simulation from which initial conditions for the SO3 simulations are taken. We have significantly changed and restructured the paragraph to improve the clarity.

Line 162-164 : Has the drift vs. the climate change forcing been quantified?

Quantifying this using typical methods would be challenging given the experimental setup we use. Typically, one would quantify drift by calculating the trend during a control run and this could be compared to trends during the later periods during anthropogenic climate change. In our setup, we have only five years of usable data, which is not enough to reliably calculate a trend. As well, the bulk of anthropogenic climate change occurs between the historical, present, and projected 5-year simulation periods rather than during them, both via the eddy-permitting transient simulation and the forcing dataset. Finally, since we do not quantify the effects of climate change as a trend, but as a difference of means, traditional conceptions of model drift as we describe it above may not be as relevant; even if significant drift is present, it can only have a minimal impact over the course of five years.

Section 2.2 : this only discusses the setup of SO3, which is fine because the reader is directed to the other setups in the previous section. But it would be interesting to summarise here the number of nodes, etc, to compare with SO3.

We have added the number of elements, nodes and vertical layers in the mesh used in the AWI-CM-1-1-MR setup, as well as the approximate size of saved 3D data.

Line 174-175 : the references on these lines have got the brackets in the wrong place, they should be around the year only.

This has been corrected.

Line 184-185 : there are two interpolations to get the forcing variables from the donor grid to the SO3 grid. This seems like it could exacerbate potential impacts, such as not conserving certain variables. Did any special care have to be taken with this process?

This was an error and has been corrected. The forcing data is on a regular grid natively, since it comes from the atmospheric component of the coupled AWI-CM-1-1-MR model. Only one interpolation has to occur and it is performed internally by FESOM. In the uncoupled setup, this is a bilinear interpolation and is not conservative. The reasoning here is that the atmosphere does not react to the ocean in the uncoupled setup, meaning atmospheric properties, particularly fluxes, which one might want to conserve, will likely already be somewhat disconnected from the evolution of the modeled ocean. In contrast, the coupled setup conserves these properties.

Line 210 : there's a rogue "e." at the beginning of the line. Is this a remnant of an old labelling scheme?

This has been corrected to match the rest of the subheadings.

Line 214 and 218 : the prime isn't the same symbol as actually used in equation (3).

The prime symbol has been changed to the one in equation (3).

Line 220 : There is currently no explanation why the EKE is coarsened to five-day means. This would be expected to reduce the EKE and may harm the comparison.

This was done to reduce the computational burden of producing figures 1 and 3. The data is coarsened to five day means after EKE is calculated and will therefore not reduce EKE. We can expect extremes to be dampened, but taking a mean of five day means will yield the same result as the mean of the daily dataset. Since we do not investigate extreme values of EKE, we do not consider this to impact the study.

We have explained that the data was coarsened to reduce the computational burden at line 248.

Line 222-223 : the averaging region is initially described as being from 45oS to 60oS and then as having the area northward of 40oS removed. Why is this necessary if the averaging starts at 45oS? And why is the Brazil/Malvinas confluence removed?

Only the Brazil/Malvinas region between 57 and 29E and north of 40S is removed.

This region is expected to respond to different climatic drivers (Beech et al., 2022) and part of this eddy-rich region lies outside of the study domain (North of 45S). For clarity and concision, we prefer to focus on a region that is, in theory, responding to relatively consistent physical drivers and is reasonably completely contained within our study region.

We have added a brief explanation of this at line 251.

Line 235 : the 50% underrepresentation of the observations would be clearer if you noted the change in y-axis between panels here.

This has been added.

Line 236-237 : is the "less Gaussian" appearance the result of a specific test or the use of the eyeball norm?

We have added tables of specific statistics in the Supplementary material.

Line 238-240 : does a deviation from normality have a physical interpretation reflecting eddy behaviour? Similarly, would multimodality or skewness reflect something physical about the circulation?

We report the deviation from normality as a precaution for the basic statistics we use; even the mean may not be an effective representative of the data if it is multimodal, for example. However, by displaying the distributions of each dataset, and now, by including the descriptive statistics in the supplementary materials, we hope that these caveats are transparent to the reader.

Skewness may reflect a trend within the dataset, either from initialization shock or the climate change signal. So however unlikely this is given our short datasets, we have modified the analysis to remove linear trends from the data before analysis in figures 1 and 3. Particularly given that any trends due to climate change would introduce a systemically larger bias to the later periods. We have added this explanation in the methods section.

Both multimodality and skew of EKE could reflect multiple modes of circulation, as in the Kuroshio region in Beech et al. (2022), or seasonality. There is no reason, as far as we are aware, to expect multiple modes of circulation affecting eddy activity in the southern ocean (Frenger et al., 2015). Therefore, we would find it somewhat distracting to include in the text. Seasonality is more likely considering the impacts of sea ice, although we expect this to have a greater impact

on the observational dataset through data availability. We have included this in the discussion at line 432.

Figure 1 caption : why "Real" magnitudes of EKE for panels a-c? In the description for panel (c) SO3 isn't fully capitalised.

Real was included to distinguish these values from EKE anomalies in the other panels. The capitalization has been corrected and real has been removed.

Line 254-256 : this short of regional discrepancy could be because of the use of EKE at depth, instead of the surface geostrophic EKE.

The AWI-CM-1 ensemble, which does use surface geostrophic velocities for EKE calculations, also does not effectively reproduce observed eddy-activity in this region, despite broadly matching the regional EKE distribution seen in SO3 and the observational dataset elsewhere. This suggests to us that the choice of velocities is not the cause of the problem. More broadly, we have addressed the use of geostrophic velocities at depth both above and in the text.

Line 280-281 : It is noted that "other factors" may contribute to SO3 not showing an increase in EKE. But what are they? If the authors have such factors in mind, listing them here for later discussion would help the reader.

This has been removed. We now discuss supplementary figures 3 and 4 which outline differences in atmospheric contributions.

Line 341 : what is "spurious" about such decline? Declines such as this could occur due to a change in the position of the mean eddy field, which would not be unexpected if the change in forcing results in a change in the mean circulation. A region of strong EKE being relocated by a few 10's of km could result in a large decrease in local EKE.

"Spurious" was intended to mean inconsistent within the ensemble based on Figure 4. So while EKE decline could feasibly occur, most of the localized regions of decline appear to be the result of natural variability rather than the climate change signal. We have changed the wording of this section to make this point clear.

Figure 5 : the individual panels haven't been labelled

This has been corrected.

Line 361 : Is -> is

This has been corrected.

Line 417 : extra bracket in the Hewitt reference.

This has been corrected.

References:

Beech, N., Rackow, T., Semmler, T., Danilov, S., Wang, Q., and Jung, T.: Long-term evolution of ocean eddy activity in a warming world, Nat. Clim. Change, 12, 910–917, https://doi.org/10.1038/s41558-022-01478-3, 2022.

Frenger, I., Münnich, M., Gruber, N., and Knutti, R.: Southern Ocean eddy phenomenology, J. Geophys. Res. Oceans, 120, 7413–7449, https://doi.org/10.1002/2015JC011047, 2015.

Hallberg, R.: Using a resolution function to regulate parameterizations of oceanic mesoscale eddy effects, Ocean Model., 72, 92–103, https://doi.org/10.1016/j.ocemod.2013.08.007, 2013.

Sein, D. V., Koldunov, N. V., Danilov, S., Wang, Q., Sidorenko, D., Fast, I., Rackow, T., Cabos, W., and Jung, T.: Ocean Modeling on a Mesh With Resolution Following the Local Rossby Radius, J. Adv. Model. Earth Syst., 9, 2601–2614, https://doi.org/10.1002/2017MS001099, 2017.